# *Hemigrapsus sanguineus* in Long Island salt marshes: experimental evaluation of the interactions between an invasive crab and resident ecosystem engineers

Bradley J. Peterson, Alexa M. Fournier, Bradley T. Furman and John M. Carroll[1]

School of Marine and Atmospheric Sciences, Stony Brook University, Stony Brook, NY, United States
[1] Current affiliation: Biology and Marine Biology, University of North Carolina Wilmington, Wilmington, NC, United States

## ABSTRACT

The invasive Asian shore crab, *Hemigrapsus sanguineus*, has recently been observed occupying salt marshes, a novel environment for this crab species. As it invades this new habitat, it is likely to interact with a number of important salt marsh species. To understand the potential effects of *H. sanguineus* on this ecosystem, interactions between this invasive crab and important salt marsh ecosystem engineers were examined. Laboratory experiments demonstrated competition for burrows between *H. sanguineus* and the native fiddler crab, *Uca pugilator*. Results indicate that *H. sanguineus* is able to displace an established fiddler crab from its burrow. Feeding experiments revealed that the presence of *H. sanguineus* has a significantly negative impact on the number as well as the biomass of ribbed mussels (*Geukensia demissa*) consumed by the green crab, *Carcinus maenas*, although this only occurred at high predator densities. In addition, when both crabs foraged together, there was a significant shift in the size of mussels consumed. These interactions suggests that *H. sanguineus* may have long-term impacts and wide-ranging negative effects on the saltmarsh ecosystem.

## INTRODUCTION

Marine systems that have suffered from a high level of human disturbance are more susceptible to successful invasion by non-indigenous species. Anthropogenic disturbances such as predator removal, land-use changes, and eutrophication may alter an environment to the extent that native species lose their locally-adapted advantages. If this happens, non-native species benefit from the disruption of ecosystems and may be able to establish themselves in the community (*Byers, 2002*). Once established, invasive species can have an array of negative effects on the communities they invade: they may compete with native species for food and other resources, introduce pathogens that infect humans or native species, reduce the recreational or commercial value of the area, and ultimately alter

Corresponding author
Bradley J. Peterson,
bradley.peterson@stonybrook.edu

biodiversity, potentially causing native species to become drastically reduced or extirpated (*Ruiz et al., 1997*). Southern New England has suffered from the invasion of many plant and animal species (e.g., the common reed, *Phragmites australis*, and the green crab, *Carcinus maenas*), some of which have significantly changed the structure of communities where they were introduced (*see Angradi, Hagan & Able, 2001* (*P. australis*); *Grosholz & Ruiz, 1996* (*C. maenas*)). One recently introduced species of concern in New England is the Asian shore crab, *Hemigrapsus sanguineus.*

The Asian shore crab is believed to have been introduced to the northwest Atlantic once or multiple times by ballast water discharged from ocean-going ships (*Epifanio et al., 1998*). The crab was first observed in the United States in 1988, in Townsend Inlet, New Jersey (*McDermott, 1991*). It was soon found in Long Island Sound and has since spread north to Maine and as far south as North Carolina (*Park, Epifanio & Grey, 2004*). *Hemigrapsus sanguineus* is now well established on the northeast coast of the United States, where densities rival or exceed those in their native Asia (*Altieri et al., 2010*), and has displaced other crab species in rocky areas to become the dominant crab in some communities (*Ahl & Moss, 1999*; *Gerard, Cerrato & Larson, 1999*; *Cassanova, 2001*).

*Hemigrapsus sanguineus* is a cryptic predator and relies on the shelter of rocks, shells, and seaweed during low tide to help prevent desiccation, predation, and thermal stress (*Brousseau & Baglivo, 2005*; *Brousseau et al., 2002*). The crab has been described as selectively inhabiting rocky areas rather than sandy coastal or estuarine habitats, seagrass beds, or marshes (*Ledesma & O'Connor, 2001*). Recently, however, it has been observed in several Long Island and Connecticut salt marshes using macroalgae (*Ascophyllum*, *Fucus*) rather than rocks for shelter (Fig. 1) (*Brousseau, Kriksciun & Baglivo, 2003*; *Fournier, 2007*). *Brousseau, Kriksciun & Baglivo (2003)* found Asian shore crabs, which are unable to excavate burrows themselves, occupying burrows dug by fiddler crabs (*U. pugnax*) at the edge of a Connecticut salt marsh, suggesting the native fiddler crab may be facilitating the invasion of *H. sanguineus* into a new environment. If *H. sanguineus* is in fact able to exploit this new environment, its range along the Atlantic coast may expand beyond its present boundaries. Concomitantly, Asian shore crab abundance could increase in areas within its current range as novel habitat becomes populated, which could have important consequences for a number of salt marsh resident species.

An important and characteristic marsh resident is the fiddler crab, *Uca* sp. Fiddler crabs have a facultative mutualism with cordgrass. In muddy marsh sediments, cordgrass roots help stabilize the mud and provide structural support for crab burrows. The burrowing behavior of crabs, in turn, facilitates the growth of cordgrass by oxygenating dense marsh soils, improving drainage, and increasing the decomposition of underground plant debris in an otherwise anoxic environment (*Montague, 1982*). In sandy sediments, the presence of fiddler crabs may also increase nutrient availability to cordgrass (*Holdredge et al., 2010*). In this way, fiddler crabs play an important role in determining the primary productivity and structure of the salt marsh and may influence the rate of marsh accretion and succession (*Bertness, 1985*). The Asian shore crab has been found to utilize burrows dug by fiddler crabs; this could cause a change in *Uca* burrowing behavior or drive fiddler crabs from

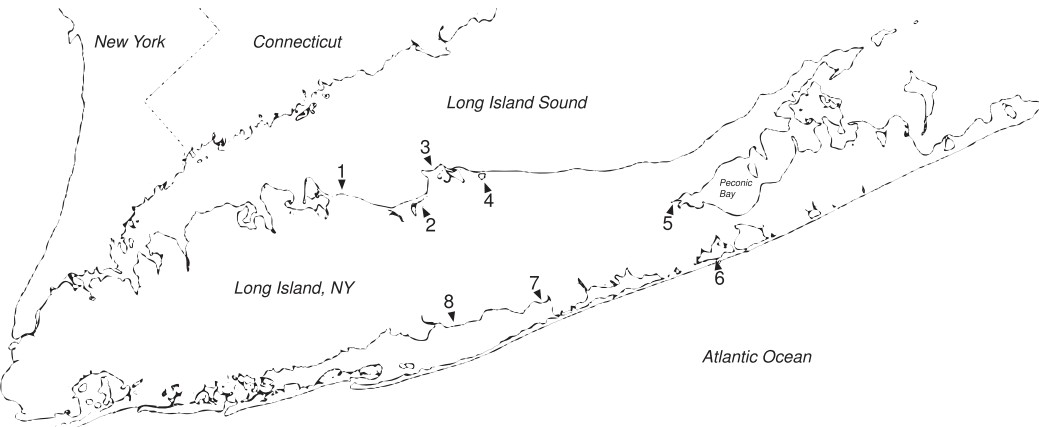

**Figure 1 Long Island salt marshes invaded by *Hemigrapsus*.** Sites of *Hemigrapsus sanguineus* presence/absence surveys on Long Island, NY. (1) Crab Meadow Beach; (2) West Meadow Creek; (3) Flax Pond; (4) Mt. Sinai Harbor; (5) Goose Creek; (6) Dune Road Marsh; and (7) Fireplace Neck.

the area. In addition, fiddler crab mating, which occurs inside burrows, may be limited in the presence of *H. sanguineus*; this also could potentially reduce *Uca* populations in areas where Asian shore crabs are present.

In New England salt marshes, another species of particular importance that might be affected by this new invasion is the ribbed mussel (*Geukensia demissa*), an important salt marsh engineer. Mussels are limited by attachment sites and benefit from the stability provided by cordgrass stems and root mass. As they filter water, ribbed mussels take up particulate nitrogen and convert it to ammonia, dissolved organic nitrogen, and semi-solid biodeposits, the bulk of which are excreted and quickly utilized by other marsh residents (*Jordan & Valiela, 1982*). Cordgrass, which is nitrogen-limited, benefits from the nutrient enrichment caused by ribbed mussels; where mussels are associated with cordgrass, the plants productivity is higher (*Bertness, 1984*). In addition, mussels produce proteinaceous byssal threads, which further stabilize the substrate and trap sediments.

Although *H. sanguineus* might not have a strong direct impact on mussel populations—they are relatively small and capable of consuming only small mussels (*Bourdeau & O'Connor, 2003*)—a more significant impact might arise with their interactions with a major salt marsh predator, the green crab (*Carcinus maenas*), a long established invader. As *H. sanguineus* comes into the marsh in greater numbers, competitive interactions between the crabs will likely increase. This may indirectly affect mussel populations; despite being a potential shared prey for both crab species, the presence of multiple predators rarely has an additive effect on prey (*Griffen, 2006*). Rather, when multiple predators forage together, risk of predation on the shared prey is often either enhanced (greater than expected) or reduced (less than expected), known as emergent effects (*Sih, Englund & Wooser, 1998*). Aggressive interactions between the two crab species could increase, shifting behavior from feeding to antagonism, and ultimately having an impact on mussel populations. Understanding if the two predators foraging together have an emergent effect
on ribbed mussels is important for fully understanding the impacts of marsh invasion by *H. sanguineus*.

The primary objectives of this study were: (1) to experimentally examine whether competition for burrows occurs between this invasive crab and the native fiddler crab, *Uca pugilator*; and (2) to investigate the emergent effects of multiple predators—the green crab, *Carcinus maenas* and the new invader, *Hemigrapsus sanguineus*—on ribbed mussel predation.

## METHODS

### Burrow competition experiment

*Hemigrapsus sanguineus* were collected by hand along the shore of Stony Brook Harbor, Long Island, NY, and maintained in an indoor saltwater aquarium, filled with rocks to provide shelter, for no more than one month prior to use in trials. Crabs were fed Wardley Algae Discs (Hartz Mountain Corp., Secaucus, NJ) supplemented with fresh algae, *Ulva lactuca.Uca pugilator* were collected by hand from Flax Pond and Scallop Pond, Long Island, NY, and kept in a tank built to mimic natural conditions: a sloping layer of sifted beach sand leading to a pool of salt water approximately 4 cm deep on one end of the tank. The crabs were able to submerge themselves, forage on the dry sand surface, or excavate burrows down to moist bottom sand. *Uca* crabs were kept no more than one month prior to use in trials and were fed a ground mixture of Wardley Algae Discs and Tetrafauna Hermit Crab Cakes (Tetra Holdings, Blacksburg, VA). Male and female crabs of both species were used, and each crab was used in only one trial. Equal number of males and females of each species were used to determine whether there was a difference based on the sex of either crab. In addition, carapace width was recorded to determine if size affected the outcome between the crabs. The average female carapace width of *U. pugilator* was $18.6 \pm 0.61$ mm and ranged in size from 17.5 to 19.2 mm. The male size range was 16.9–21.2 mm and averaged $19.41 \pm 1.32$ mm. In comparison, the average size of *H. sanguineus* females was $15.8 \pm 2.0$ mm and ranged in size from 12.0 to 19.2 mm. The average male sizes were $17.9 \pm 2.4$ mm and ranged from 13.7 to 21.2 mm.

For each experiment, an arena was set up in a 19-l bucket filled with damp beach sand, sifted with 1 mm mesh, to a depth of approximately 20 cm and smoothed flat, resulting in a $\sim$616 cm$^2$ surface area arena. Near the arena edge, an artificial burrow 2.3 cm in diameter and 6 cm deep was made. A 125 W heat lamp was positioned 22 cm from the sand surface, providing thermal stress to promote the fiddler crab's burrowing response and increasing desiccation stress for *H. sanguineus* (Fig. 2). During the experimental runs, the temperature at the sand surface averaged 27 °C while the temperature inside the burrow averaged 21 °C, thus reflecting normal summer temperatures of the natural habitat. One randomly selected fiddler crab was placed in the arena and given sufficient time to occupy the burrow. Then a randomly selected *H. sanguineus* was added opposite the burrow. After 20 min, the burrow occupant was recorded, and any new burrowing activity was noted. Video recordings were made of eighteen trials to examine the behavior of the crabs during the experiment.

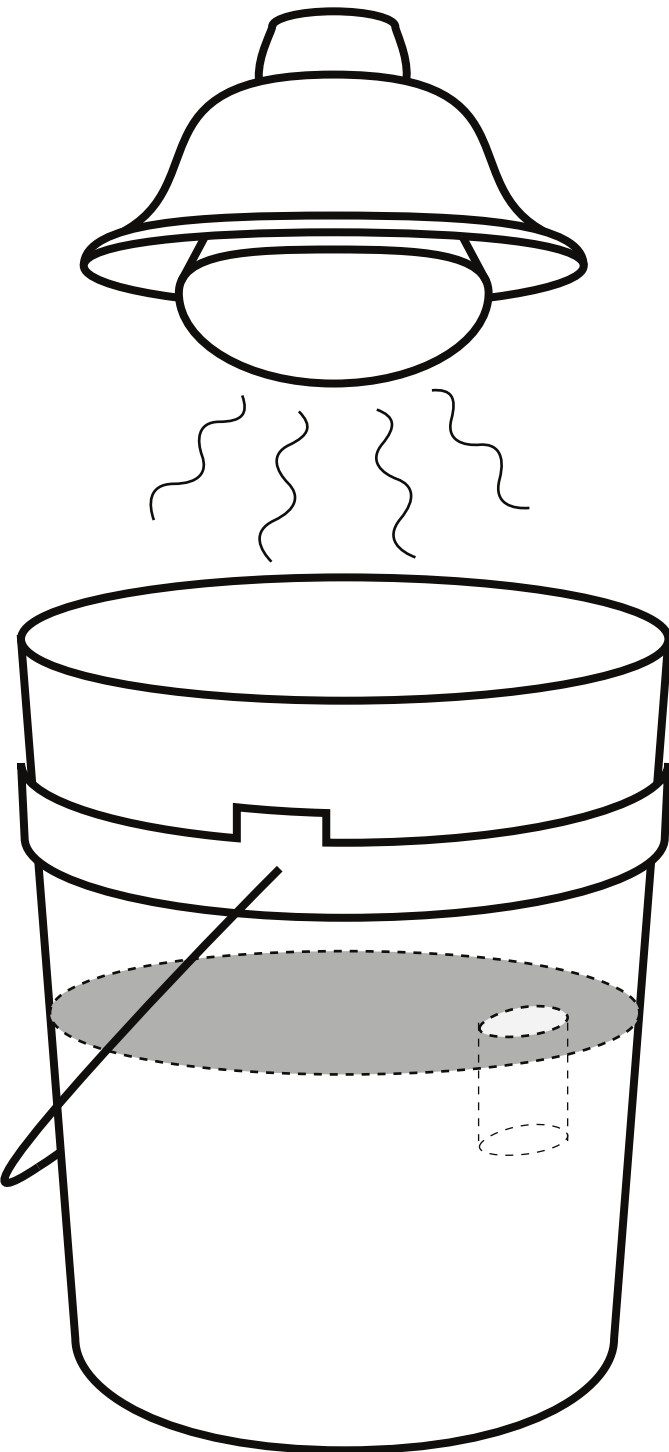

**Figure 2 Arena set-up for burrow competition experiment.** Arena set-up for the burrow competition experiment, showing the 19 l bucket and the position of the heat lamp.

To compare the competitive interactions of *U. pugilator* with conspecifics to its behavior with the invasive crab, additional trials were performed as above, but with a second *U. pugilator* in the place of *H. sanguineus*. Crabs were labeled with small (5 mm square) vinyl number tags, attached to the carapace with cyanoacrylate glue. This was done to provide a means of distinguishing the original crab from the one added. These experiments were conducted in such a way to compare differences of the success of *H. sanguineus* in possessing a burrow based on crab size (CW) and sex of the *U. pugilator* defender.

To control for laboratory artifacts, these experiments were conducted in the field as well. Experiments were performed at the beach on Scallop Pond, Southampton NY using resident fiddler crabs and *H. sanguineus* collected from Old Ponquogue Marine Park, Southampton NY. Arenas were created using the top half of a 19-l bucket driven into the sand approximately 3–5 cm. As in the laboratory experiments, an artificial burrow 2.3 cm in diameter and 6 cm deep was made in the arena. One randomly selected fiddler crab was placed in the arena and given sufficient time to occupy the burrow, then a randomly selected *H. sanguineus* was added opposite the burrow. After 20 min, the burrow occupant was recorded and any new burrowing activity noted. Conspecific trials, using two *U. pugilator* crabs, were performed in the same fashion.

## Mussel feeding experiment

To determine the emergent effects of *H. sanguineus* and the green crab, *Carcinus maenas*, on ribbed mussels, a feeding experiment was performed using a fixed number of ribbed mussels and 6 different combinations of *H. sanguineus* and *C. maenas* crabs. The treatments were (i) one *H. sanguineus*; (ii) one *C. maenas*; and (iii) one *H. sanguineus* plus one *C. maenas*; (iv) two *H. sanguineus*; (v) two *C. maenas*; and (vi) two *H. sanguineus* plus two *C. maenas* (1Cm, 1Hs, 1Cm + 1Hs; 2Cm, 2Hs, 2Cm + 2Hs). The use of these different density combinations allowed both the impact of increasing density of conspecifics on mussel consumption as well as interspecific effects. *Hemigrapsus sanguineus* and *Carcinus maenas* were collected from Old Ponquogue Bridge Marine Park, Southampton, NY. Additional *C. maenas* were collected by trawl within Shinnecock Bay (NY). Sizes of *H. sanguineus* ranged from 18 to 24 mm carapace width (CW), with an average CW of 21.0 mm, while the sizes of *C. maenas* ranged from 53 to 76 mm CW and averaged 62.1 mm CW, within sizes commonly observed in marine habitats around Long Island and similar to sizes used in other studies (*Griffen, 2006*; *Griffen, Guy & Buck, 2008*). Soft-shelled, gravid females and crabs with missing or damaged chelipeds were not used in the experiment. All crabs were starved 24 h prior to use in each experiment.

A large clump of approximately 800 individual ribbed mussels (*Geukensia demissa*) was collected from Red Creek Pond, Southampton, NY. The mussels were measured and classed by size, from <1.0 to 6.5 cm in 0.5 cm increments, then counted. This was done to utilize the natural size distribution of ribbed mussels in the field (Fig. S1). The mussels were then divided into 18 identical sets of 39 mussels each, reflecting the natural size distribution observed. Each set was allowed to attach via byssal threads to 12.5 cm plexiglass circles in a flow-through seawater system for 48 h before being used in

experimental trials. Prior to the experiment, mussels were able to feed on phytoplankton present in the unfiltered seawater but were not given additional food.

The feeding experiments were performed at the Stony Brook-Southampton Marine Lab. Each of the 6 experimental treatments had 3 replicates; all were run in 18 separate polypropylene containers (33 cm × 20 cm × 12 cm deep) haphazardly arranged in a 3-tiered, flow-through sea table with unfiltered seawater. Water was directed into each individual container, and allowed to flow over the top for water exchange. The water level in the sea table was kept at 9 cm deep, enough to help maintain constant temperature but preventing water exchange between different containers, thus keeping them independent (Fig. 3). All crabs were measured and sexed before use. Only adult males of both species were used in these experiments. To begin, a pre-made mussel clump was placed in each container, and then the randomly-assigned crab treatment was added. The experiment was run for 48 h.

After completion of the experimental run, total prey consumption was measured by removing the mussels from each bin and counting how many remained from each size class. The consumed mussel count was converted into an approximate biomass value for each bin and treatment using a regression equation,

$$y = 0.025x^{2.16}$$

where $x$ = mussel shell length in centimeters and $y$ = whole tissue dry weight in grams ($p < 0.001$; $r^2 = 0.97$; Fig. 4). This relationship was calculated by collecting tissue dry weights of 50 individual mussels from each size class and regressing biomass (dry weight) to shell length.

## STATISTICAL ANALYSIS

In total, this study consisted of two manipulative experiments: a burrow competition experiment and a mussel feeding experiment. Results in all comparisons were considered to be significant when $p < 0.05$.

For the burrow competition experiment, statistical comparisons of the final burrow occupant were made via Chi-square tests (Sokal & Rohlf, 2000; Zar, 1999). To determine whether an emergent multiple predator effect occurred, log transformed prey survivorship data was analyzed with a two-way ANOVA with the presence/absence of each predator species treated as a separate factor (Sih, Englund & Wooser, 1998; Griffen, 2006). A significant interaction term between the two factors indicates the presence of an emergent effect of combining the two predator species. In addition, we compared the observed proportion of mussels consumed to values predicted by the multiplicative risk model (Sih, Englund & Wooser, 1998; Wong, Peterson & Kay, 2010):

$$E_{C,H} = P_c + P_H - (P_C P_H)$$

where $E_{C,H}$ is the predicted proportion of the mussels consumed by Carcinus and Hemigrapsus when foraging together, $P_C$ is the observed proportion of mussels consumed by Carcinus in isolation and $P_H$ is the observed proportion of mussels consumed by

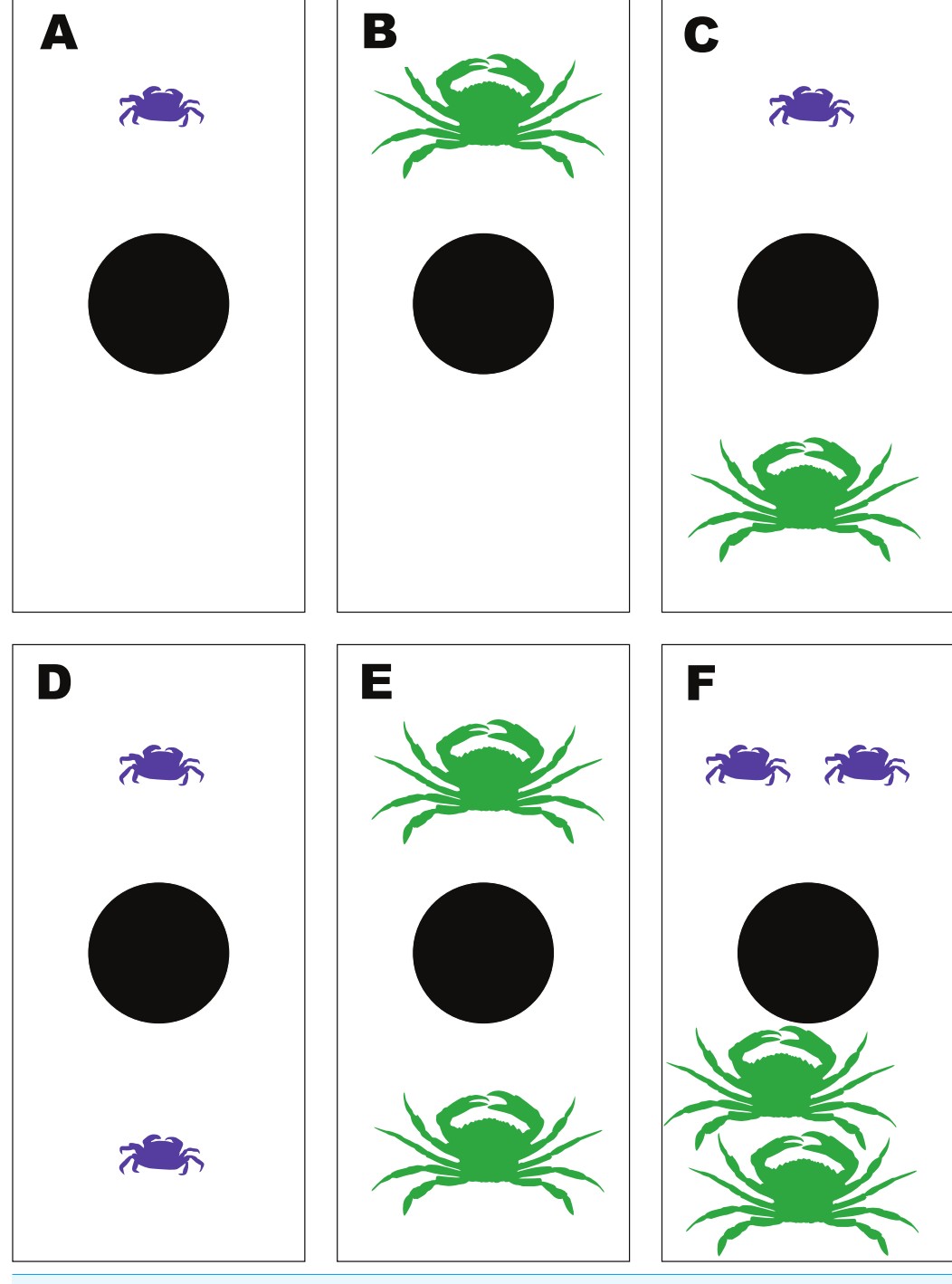

**Figure 3 Experimental set-up for the mussel predation experiment.** Diagram of the experimental set-up for the multiple predator experiments between *Hemigrapsus sanguineus* and *Carcinus maenas* feeding on shared mussel prey, illustrating the 6 different treatment combinations—(A) 1Hs; (B) 1Cm; (C) 1Hs + 1Cm; (D) 2Hs; (E) 2Cm; (F) 2Hs + 2Cm.

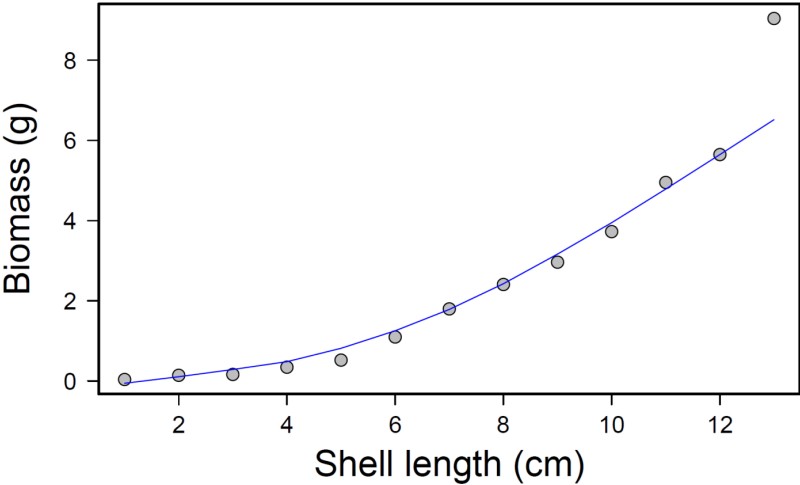

**Figure 4 Mussel biomass vs. shell length regression.** Mussel biomass vs. shell length regression; equation: $y = 0.025x^{2.1561}$, where $y$ is the tissue dry weight in g and $x =$ mussel shell length in cm.

*Hemigrapsus* in isolation. Transformations were not necessary to meet assumptions of normality. The multiplicative risk model is used to predict how many mussels should be eaten by both predators together based on individual foraging rates, and therefore divergence from expected also illustrates an emergent effect, and also the direction of the effect (risk enhancement, risk reduction). $P_C P_H$ is a correction factor for mussels that can't be consumed twice. Both the two-way interaction approach and the multiplicative risk model have been used separately in previous studies examining multiple predator affects. We chose to use both approaches to identify whether an emergent effect occurred and determine the direction of the effect. These statistical analyses were conducted using the Sigma Plot 11.0 statistical software package.

## RESULTS

### Burrow competition experiment

There was a statistically significant difference in the crab species occupying the burrow after 20 min ($p < 0.001$). After a total of 38 trials, the burrow was occupied by *H. sanguineus* alone 14 times, by *Uca pugilator* alone 4 times, and by both crabs 20 times. In no trial was the burrow unoccupied at the end of 20 min. The success of *H. sanguineus* in the trials was not dependent upon the size (CW) ($p = 0.545$), nor sex ($p = 0.633$) of *U. pugilator*. Neither was it dependent on the size (CW) ($p = 0.592$), nor sex ($p = 0.504$) of *H. sanguineus*.

The behavior of *U. pugilator* in response to *H. sanguineus* varied from trial to trial. On some occasions, *H. sanguineus* would simply enter the burrow occupied by *U. pugilator* and both crabs would remain together for the length of the experiment without any outward displays of aggression. Other times, *U. pugilator* would leave the burrow upon entry of *H. sanguineus*, then actively attempt to re-take it using aggressive behaviors and territorial displays such as acoustical sound production by vibration of the major chelae (in males) and repeated approaches to the burrow (both sexes). On two occasions, *U. pugilator*
males were observed using their dominant claw to pry *H. sanguineus* out of the burrow. In two trials, *U. pugilator* excavated a new burrow after being displaced from the original burrow by *H. sanguineus*.

The behavior of *H. sanguineus* would also be aggressive when *U. pugilator* defended or attempted to re-occupy the burrow. *H. sanguineus* would often approach an occupied burrow with claws raised; if already inside the burrow, it could be seen stretching its chelipeds to occupy as much space as possible. On several occasions, it was observed that *H. sanguineus* would use its claws as pinchers to deflect attempts by *U. pugilator* to re-enter the burrow.

In the outdoor trials with *H. sanguineus* and *U. pugilator*, the same behaviors witnessed in the lab trials were observed in the field. In 20 trials, the burrow was occupied by *H. sanguineus* 11 times (55%), by *U. pugilator* 5 times (25%), and by both crabs 4 times (20%). These results of the field trials were not significantly different from those in the laboratory ($p = 0.40$).

### Mussel feeding experiment

When foraging alone, *Carcinus maenas* consumed 17% of the mussels offered to them, while only 8% were consumed by *Hemigrapsus sanguineus* foraging alone, likely due to a size threshold of the prey. When one of each species was present together, the combined consumption was 14%. However, the proportion of mussels consumed did not vary significantly across predator treatments at low predator density ($p = 0.801$). Increasing from single predators to conspecific pairs of *C. maenas* more than doubled the number of mussel prey consumed (62% consumed), but had no effect on *H. sanguineus* (5% consumed; Fig. 5). Two *C. maenas* together consumed significantly more mussels than both two *H. sanguineus* and the mixed treatment with two of each species ($p < 0.001$ for both).

There was not a significant emergent effect on prey survival when single individuals of the two predator species foraged together based on the additive experimental design (*C. maenas* × *H. sanguineus* interaction, $p = 0.294$; Table 1, Figs. 5 and 6). The lack of an emergent effect was confirmed by comparing our observed values to those predicted by the multiplicative risk model ($p = 0.646$; Table 2). When applying the additive design to high predator densities (2Cm, 2Hs, 2Cm + 2Hs), there was a significant interaction, suggesting an emergent effect ($p < 0.001$; Table 1). This emergent effect was also apparent when comparing the actual consumption observed to that predicted by the multiplicative risk model ($p < 0.001$; Table 2, Figs. 5 and 6).

In addition to a change in the number of mussels consumed, the mean size of ribbed mussels eaten shifted from larger to smaller mussels when both predators foraged together (Fig. 7). When foraging alone, *C. maenas* consumed mussels ranging from the smallest size class ($<1.0$ cm) to a length of 5.5 cm, while *H. sanguineus* consumed the smallest size range of mussels, from $<1.0$ to 2.0 cm in length. In treatments when both predators foraged together, the lack of consumption on the larger mussel size classes was dramatic (Fig. 7). A Mann–Whitney rank sum test was performed on the data and showed that the median

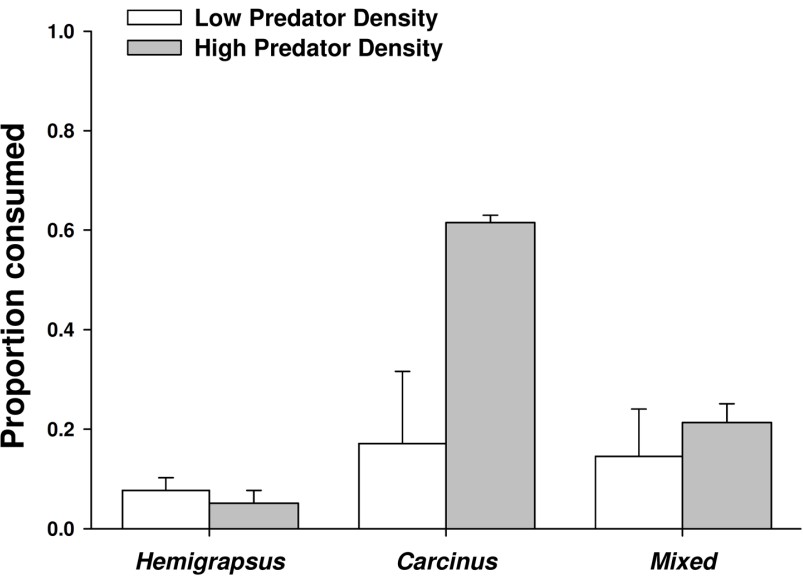

**Figure 5 Crab predation of mussels as individuals and heterospecific and conspecific pairs.** Proportion of mussel prey consumed (mean ± SE, $n = 3$) foraging as single individuals and in heterospecific and conspecific pairs. Low predator density refers to one individual or one of each in the mixed trial. High predator density refers to two individuals or two of each in the mixed trial.

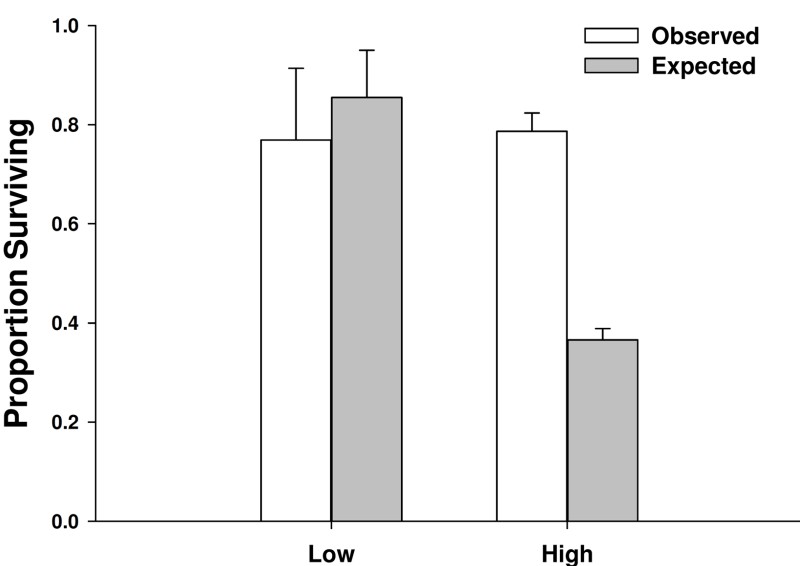

**Figure 6 Proportion of mussels surviving in low vs. high crab densities.** Observed and expected mussel prey survival when predators foraged together at low (one of each species) and high (two of each species) predator densities (mean ± SE, $n = 3$). Expected survival was calculated using the multiplicative risk model $E_{C,H} = P_C + P_H - (P_C P_H)$, where $P_C$ is the proportion of the mussels consumed by *Carcinus maenas* and $P_H$ is the proportion of mussels consumed by *Hemigrapsus sanguineus* when foraging along. Low predator densities are the observed and expected survival of mussels when 1 individual of each species foraged together. High predator densities are the observed and expected survival of mussels when 2 individuals of each species foraged together. * when $p < 0.05$.

**Table 1 ANOVA Table of the mussel predation experiment.** Results of ANOVAs used to compare observed predation by Carcinus maenas and Hemigrapsus sanguineus to expected predation based on the additive design and method described by *Griffen (2006)*. Crab predation was measured as the proportion of mussels consumed over the 48 h experiments in the presence/absence of each predator.

| Source | df | SS | F | P |
|---|---|---|---|---|
| **Test of additive for low density (two-way ANOVA)** | | | | |
| *C. maenas* | 1 | 0.0430 | 1.858 | 0.210 |
| *H. sanguineus* | 1 | 0.00197 | 0.0853 | 0.778 |
| *C. maenas × H. sanguineus* | 1 | 0.00789 | 0.341 | 0.575 |
| Error | 8 | 0.185 | | |
| **Test of additive design for high density (two-way ANOVA)** | | | | |
| *C. maenas* | 1 | 0.444 | 253.125 | <0.001 |
| *H. sanguineus* | 1 | 0.0877 | 50.000 | <0.001 |
| *C. maenas × H. sanguineus* | 1 | 0.148 | 84.500 | <0.001 |
| Error | 8 | 0.0140 | | |

**Table 2 ANOVA Table of the multiplicative risk model.** Results of One-Way ANOVAs used to compare observed predation by Carcinus maenas and Hemigrapsus sanguineus together to expected predation calculated by using the multiplicative risk model as described by *Wong, Peterson & Kay (2010)*. Predation was measured as the proportion of mussels consumed over 48 h experiments.

| Source | df | SS | F | P |
|---|---|---|---|---|
| **Low predator density** | | | | |
| Between groups | 1 | 0.0110 | 0.245 | 0.646 |
| Error | 4 | 0.180 | | |
| **High predator density** | | | | |
| Between Groups | 1 | 0.266 | 92.663 | <0.001 |
| Error | 4 | 0.0115 | 0.00287 | |

values for the two groups (*C. maenas* alone and with *H. sanguineus*) were significantly different ($p < 0.001$), with smaller mussels being consumed in heterospecific groupings. While this was affected by the increased consumption of the smaller size classes by *H. sanguineus*, it also demonstrates that *C. maenas* did not consume the larger individuals when *H. sanguineus* was present.

Finally, when considering the impact of mussel predation on removing biomass instead of prey density, the effects of both predators foraging together are even more dramatic. High density predator treatments of *C. maenas* consumed a significantly higher biomass of ribbed mussels than the low density *C. maenas* treatment. When *H. sanguineus* was present there was significantly less biomass consumed than when *C. maenas* was alone at the highest densities ($p = 0.002$, Fig. 8).

## DISCUSSION

The purpose of this study was to examine the potential impact an invading crab may have on two ecosystem engineers within the salt marsh community. The Asian shore crab,

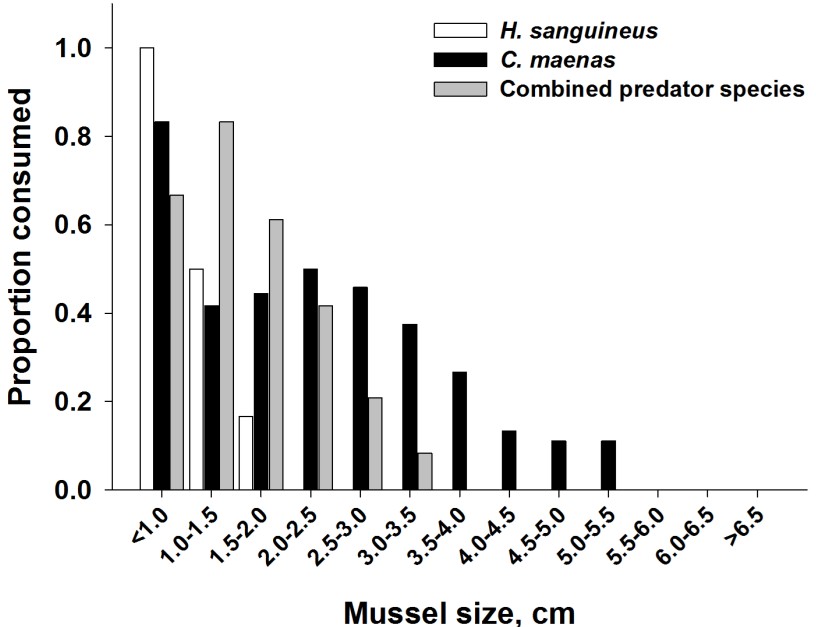

**Figure 7 Impact of heterospecific predators on reducing the size of mussels being consumed.** Proportion of mussels consumed in each size class for each predator alone and when foraging together. Prey consumption by both predator densities were combined. Shell length was measured across the longest shell axis.

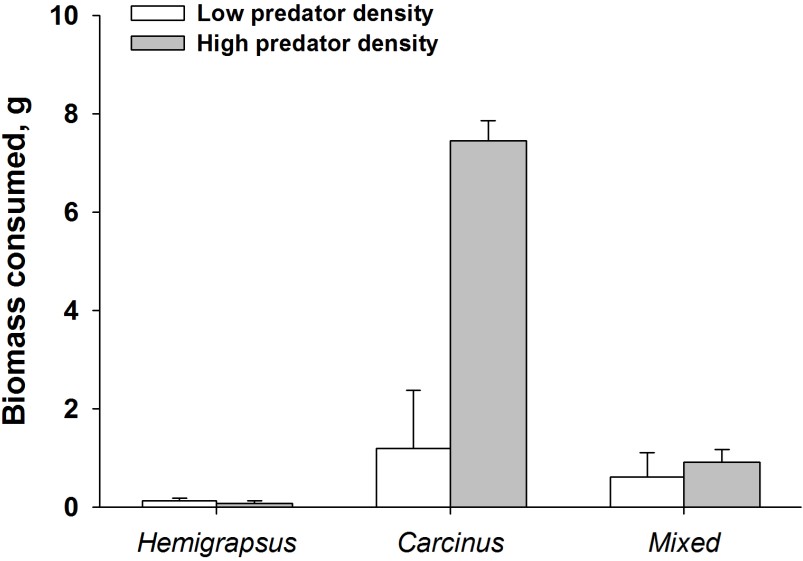

**Figure 8 Dramatic increase in mussel biomass consumed by *Carcinus maenas* at high predator densities.** Mussel biomass consumed by both predators individually and when foraging together (mean ± SE). Low predator density refers to one individual or one of each in the mixed trial. High predator density refers to two individuals or two of each in the mixed trial.

*Hemigrapsus sanguineus*, is now common on rocky shores from Virginia to Maine and has recently begun to invade salt marsh ecosystems in New York (*Fournier, 2007*) and Connecticut (*Brousseau, Kriksciun & Baglivo, 2003*). This can have serious consequences for the salt marsh ecosystem through both direct and indirect interactions. Burrow competition experiments showed that the shore crabs can utilize fiddler crab burrows and are capable of displacing the native species. Additionally, the presence of *H. sanguineus* can significantly affect the amount, size, and biomass of mussels consumed by an important marsh predator.

The burrow competition experiments performed in this study revealed that *H. sanguineus* will utilize burrows created by *U. pugilator* and is able to displace an established crab from its burrow. In the trials, *H. sanguineus* occupied the burrow alone or with *U. pugilator* significantly more often (89% of the time) than did *U. pugilator* alone (11% of the time). The outcome of these competition arena experiments were similar with respect to *H. sanguineus* occupancy in both the lab and in the field experiments (80% of field burrows were occupied by *H. sanguineus*). However, how the occupancy occurred was slightly different. In the lab arenas, *H. sanguineus* occupied the burrow alone ∼37% of the time, while it co-occupied the burrow in ∼53% of the trials. In the field, this pattern was almost opposite, which *H. sanguineus* in 55% of the burrows alone compared to sharing the burrow 20% of the time. While it is unclear why the lab and field arenas differed in how the burrow occupancy occurred, possibly due to behavioral differences between lab adapted and resident crabs used in the different experiments, the results show that *H. sanguineus* is able to use burrows created by *U. pugilator*. Therefore, fiddler crabs (*Uca pugilator* and *U. pugnax*) may act as facilitators to *H. sanguineus* invasion by providing a suitable desiccation refuge for the invaders.

It is not known what impact, if any, the presence of *H. sanguineus* will have on *Uca* sp. However, the presence of *H. sanguineus* in burrows may cause an emigration of *Uca* sp. from areas within the marsh, reduce the density of burrows, or interfere with reproductive activity known to occur within burrows. Any of these scenarios could negatively impact the salt marsh, as the importance of fiddler crab burrows to salt marsh functioning is well documented (*see Montague, 1982*; *Bertness, 1985*). However, if the benefit of *Uca* sp. to saltmarsh grasses is via nutrient deposition, as is the case on sandy shores (*Holdredge et al., 2010*), it is possible *H. sanguineus* could replace this function assuming suitable habitats remain to prevent dessication. Regardless, any potential disruptions to the mutualistic fiddler crab—cordgrass relationship are reasons for concern.

In addition, the feeding experiments suggested the possibility of an interaction between the competing predators. When one of each predator was present, there was no emergent predator effect, suggesting that the predators do not interfere with each other. However, when two of each predator was present, there was a significant emergent effect, leading to a risk reduction of their shared prey (*Sih, Englund & Wooser, 1998*). While we were unable to use the substitutive design at high predator densities, and thus cannot eliminate the possibility that the emergent effect was due to predator density in general (*Griffen, 2006*), the observed effect is likely due to interactions between the species and not within species.
First, video recordings of the two different crabs interacting indicate a reduction in feeding by *C. maenas* which was likely due to an increased amount of time spent interacting with the smaller shore crabs and second, when both crabs were present, the size class consumed shifted toward smaller mussels, a behavior which was not observed when green crabs were housed together. This experiment suggests that an emergent effect in numbers or proportion of prey consumed is not necessarily the only indicator of a multiple predator effect.

During this experiment, no traditional emergent effect was observed when individual *C. maenas* and *H. sanguineus* were foraging together, that is, the proportion of mussels consumed was as expected based on foraging of the individual crabs alone. Closer investigation of which mussels were consumed, however, did suggest a potential emergent effect. The size range of mussels consumed by green crabs were from <1 mm to 5.5 mm. However, when *C. maenas* and *H. sanguineus* foraged together, the largest mussel consumed was in the 3.0–3.5 mm size class and the bulk of the consumption was between the <1 and 1.5–2 mm size classes; therefore, despite the expected amount of consumption when both foraged together, the significant and dramatic shift in the size classes consumed suggest an emergent effect and can have important ecological consequences. These results at the very least suggest looking beyond the typical "proportion consumed" for investigating multiple predator effects.

These results are consistent with established foraging ecology of these two crab species on other mussel prey (*Mytilus edulis*; *Lohrer & Whitlatch, 2002*; *Griffen, 2006*). In this study, the observed emergent effect of the two predators when foraging together was dependent on the predator density—only when two of each predator were foraging together was an effect observed. Prey survival was greater than expected based on the additive experimental design at high predator densities. The detection of risk reduction with the additive experimental design implies that predation interference occurred between the two species, decreasing predation rates below those observed when individuals of each species foraged independently. The shift in prey size structure consumed by the predators when foraging together is likely another indication of interference. This may have a positive impact on the larger size classes of *Guekensia demissa*. Due to the competitive interference between the two species, it is likely that *C. maenas* might not be afforded time to be a selective feeder, shifting its 'preferred' prey size down to smaller size classes in presence of interspecifics (*Wong, Peterson & Kay, 2010*).

*Hemigrapsus sanguineus* consumed a greater number of smaller ribbed mussels than did *C. maenas*, likely because of its smaller size. Within existing populations of ribbed mussels, the largest individuals should benefit from the reduced predation pressure from *C. maenas* as salt marsh populations of *H. sanguineus* increase. However, both smaller mussels and mussel recruitment will likely be negatively impacted over time. Experimental treatments of *H. sanguineus* consumed more mussels in <1.0 cm size range than either of the other treatments, an observation also made in other studies (*Bourdeau & O'Connor, 2003*). As densities of *H. sanguineus* increase, wild populations of ribbed mussels within this size class will be most affected. This can have especially dire consequences for mussel

populations—as *H. sanguineus* populations rise to replace the biomass of predators they are displacing (*O'Connor et al., 2008*; *Hudson, 2011*), they can replace the feeding pressure exerted by *C. maenas*. However, this feeding pressure would be specifically targeting the recruiting classes of mussels, potentially leading to recruitment failure and population decline (*O'Connor et al., 2008*).

Previous surveys show that *Hemigrapsus sanguineus* is present in salt marshes throughout coastal Long Island (*Fournier, 2007*), and these experiments demonstrate that *H. sanguineus* interacts with two of the most important marsh engineers in ways that may have detrimental effects on the ecosystem. While densities of *H. sanguineus* in the salt marsh at present appear low, it is apparent that these crabs have found a habitable niche in this environment and are reproducing there (A Fournier, pers. obs., 2006). The ecological consequences of this for the salt marshes of Long Island and elsewhere are as of yet unknown.

## ACKNOWLEDGEMENTS
Many people provided field and laboratory support, including C Blachly, K  Rountos, C Wickel, M Lindsay and J Pan. Many thanks also goes to Dr. RM Cerrato, who first pointed out *Hemigrapsus sanguineus* at Flax Pond, a small salt marsh tidal creek on the north shore of Long Island, and who has been a great source of ideas and assistance with this project. The authors would also like to thank Dr. BR Silliman for reviewing and improving the manuscript.

### Funding
These experiments were funded through the Long Island Sound Study and by the Southampton Coastal and Estuarine Research Program. The funders had no role in study design, data collection and analysis, decision to publish, or preparation of the manuscript.

### Grant Disclosures
The following grant information was disclosed by the authors:
Long Island Sound Study.
Southampton Coastal and Estuarine Research Program.

### Competing Interests
The authors declare there are no competing interests.

### Author Contributions
- Bradley J. Peterson conceived and designed the experiments, analyzed the data, contributed reagents/materials/analysis tools, wrote the paper, prepared figures and/or tables, reviewed drafts of the paper.
- Alexa M. Fournier conceived and designed the experiments, performed the experiments, wrote the paper, prepared figures and/or tables, reviewed drafts of the paper.

- Bradley T. Furman and John M. Carroll analyzed the data, contributed reagents/materials/analysis tools, wrote the paper, prepared figures and/or tables, reviewed drafts of the paper.

## Supplemental Information

Supplemental information for this article can be found online at http://dx.doi.org/10.7717/peerj.472.

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
