# Peer review of "Hemigrapsus sanguineus in Long Island salt marshes: experimental evaluation of the interactions between an invasive crab and resident ecosystem engineers"

_PeerJ, doi:10.7717/peerj.472_

## Round 0.1 · original submission · Minor Revisions

The authors, thank you for submitting your manuscript. Please could you address where appropriate some of the minor revisions from the reviewers.

·

Basic reporting

This article was well-written, interesting and provides useful information for further explorations of the effects of Asian shore crabs on salt marsh systems. The mussel feeding experiments provide interesting insights into the nature of predator competition.

Experimental design

The burrow competition experiment would be clarified by providing more information on the size of the arena (area/circumference). It would also be of interest what the natural density of burrows in these areas is and if there are perhaps unused burrows that might complicate these interactions. More information on the comparisons of crab size and sex would also clarify the study.

An image of the mussel feeding experiment set up might be helpful. Were the smaller containers haphazardly arranged? Is it possible that chemical cues from other treatments altered predation rates? The results seem rather definitive but unintuitive. Many crabs happily eat each other, and there is some indication that crabs perceive biomass as an indicator of size (Hill and Weissburg 2013). If chemical cues from multiple treatments were transferred it is possible green crab foraging was suppressed as a result of perceiving many conspecifics/potential predators. This maybe a non-issue but a better representation of the set-up would clarify potential interactions between treatments.

Validity of the findings

Conclusions are well supported. Though replication for the mussel feeding experiments is low the results seem to be fairly strong across the metrics measured.

·

Basic reporting

Overall, I felt this manuscript almost all of these criteria. I did think the authors needed to provide more detail for why they used the multiplicative risk model and how the reader should interpret the values that are generated by it (currently, they simply refer to the reader to the citations) as the introduction to this approach seemed overly sparse. And, I also had a few issues with the tables and figures that I believe need to be addressed prior to publication. Specifically:

1. In Tables 1 and 2, I do not know what the response variable is that is being analyzed by the ANOVA models. All that is stated in the table caption is "crab predation" and I am unsure if they analyzed the effect of each crab species on the total number of mussels consumed, total biomass of mussels consumed, etc... This detail is essential to the reader being able to interpret the meaning of this table. In short, the table captions need to include more detail.

2. Figure 1- there are more sites listed on the map than are identified in the figure caption

3. Figure 2. what is "whole dry weight in gm"? do you mean the dry tissue weight of mussels in grams? And, are there error bars associated with these data points or are they so small that they are not visible on this scale?

4. Figure 4. I cannot understand why there are only 4 columns plotted and what is plotted on the x-axis from the information provided in this figure and figure legend. As mentioned above, the authors need to provide more information in the text for why they used the multiplicative risk model and detail for how they used it to calculate the expected survival values.

Experimental design

I felt this manuscript met all of these criteria. I did not have issues with the design of their experiments.

I would like to see the authors motivate why they used green crabs of this size, however. From my own observations, juvenile green crabs- individuals that are <10mm in carapace width- seem far more common in the salt marsh than the larger-sized individuals (53-76mm) they used in this study, but I have never been to marshes in this Long Island region where Hemigrapsus are interacting with green crabs. Given how much of the interpretation of their results hinges upon the dominant effect of this crab species, I think it is important that they justify why they used individuals of this size class.

Validity of the findings

I thought that the authors mis-interpreted the patterns in their burrow competition experiments by claiming that the results from the lab and field experiments were consistent and showed the same pattern. In the lab experiment, the most common outcome was that both crabs would occupy the burrow, while in the field experiment, co-occupancy was the least common outcome. I would like to see the authors re-phrase their interpretation of these results and provide a sentence or two in the discussion for why their results from the lab and field experiment were inconsistent.

Additional comments

Overall, I found this to be a nice study, well-written and likely important to the ecology of salt marshes along the northeastern coast of the US that are experiencing the dual invasion of these two predators. As described above, I think the figures, and in particular, their captions, need to revised prior to publication to include information that is necessary for the reader to interpret what information they are presenting.

·

Basic reporting

This paper is very well written and the figures are straightforward and well presented. It is a good “clean” manuscript. The topic is interesting, involving the interactions between a newly invading crab, the Asian Shore Crab, and two resident species, the Fiddler Crab and the Green Crab. Such new invasions provide valuable opportunities for studying mechanisms of biological invasion. The authors have made good use of this situation by proposing and then testing two basic hypotheses regarding possible interactions of these two crab species. The first hypothesis dealt with whether or not the Asian Shore Crab, which cannot itself dig burrows as protection from desiccation, would use burrows dug by Fiddler Crabs and even aggressively take over the burrows. The second hypothesis category dealt with whether or not the presence of the invading crabs would influence the feeding behaviour of the Green Crab on Ribbed Mussels. The mussels are also a prey item of the Asian Shore Crab. The reasons for studying the two types of possible interaction were well described and well related to the literature on such interactions.
In line 44, I would use “extirpated” rather than “extinct”.
In line 220, I would insert “the” before “Sigma Plot 11.0”.

Experimental design

As I have suggested above, I found the relevance and definition of the research questions to be very well described. The methods for both the “Burrow Competition Experiment” and the “Mussel Feeding Experiment” were well described and the methods allowed for rigorous statistical analysis. The burrow experiments were performed in both an indoor aquarium and in the field, to test for possible laboratory artefacts. The mussel feeding experiments were performed at both a low crab density and a higher crab density. The use of the latter density revealed interesting “emergent” feeding behaviour of the Green Crab in the presence of the Asian Shore Crab.

Validity of the findings

I think the conclusions are both interesting and valid. They are valid in the sense of the actual experimental methods being well designed to test the relevant hypotheses and being given credence by solid statistical analysis. The conclusions are well described and related to the biological invasion literature. Having done some work on Green Crabs myself, I was intrigued that these crabs, well known for their aggressive behaviour and called “le crabe enragé” in French, were so affected by the smaller Asian Shore Crab (when both species were at the “high density level”), that they ate fewer mussels and the mussels they did eat were significantly smaller in size! And that other workers using Mytilus edulis as the prey observed similar results for the two crab species.

---

## Round 0.2 · accepted · Accept

Dear Authors,

Thank you for submitting your revised manuscript.